# Boosting Semantic Segmentation by Conditioning the Backbone with Semantic Boundaries

**DOI:** 10.3390/s23156980

**Published:** 2023-08-06

**Authors:** Haruya Ishikawa, Yoshimitsu Aoki

**Affiliations:** Department of Electronics and Electrical Engineering, Facility of Science and Technology, Keio University, 3-14-1, Hiyoshi, Kohoku-ku, Yokohama 223-8522, Japan; aoki@elec.keio.ac.jp

**Keywords:** semantic segmentation, semantic boundary detection, multi-task learning

## Abstract

In this paper, we propose the Semantic-Boundary-Conditioned Backbone (SBCB) framework, an effective approach to enhancing semantic segmentation performance, particularly around mask boundaries, while maintaining compatibility with various segmentation architectures. Our objective is to improve existing models by leveraging semantic boundary information as an auxiliary task. The SBCB framework incorporates a complementary semantic boundary detection (SBD) task with a multi-task learning approach. It enhances the segmentation backbone without introducing additional parameters during inference or relying on independent post-processing modules. The SBD head utilizes multi-scale features from the backbone, learning low-level features in early stages and understanding high-level semantics in later stages. This complements common semantic segmentation architectures, where features from later stages are used for classification. Extensive evaluations using popular segmentation heads and backbones demonstrate the effectiveness of the SBCB. It leads to an average improvement of 1.2% in IoU and a 2.6% gain in the boundary F-score on the Cityscapes dataset. The SBCB framework also improves over- and under-segmentation characteristics. Furthermore, the SBCB adapts well to customized backbones and emerging vision transformer models, consistently achieving superior performance. In summary, the SBCB framework significantly boosts segmentation performance, especially around boundaries, without introducing complexity to the models. Leveraging the SBD task as an auxiliary objective, our approach demonstrates consistent improvements on various benchmarks, confirming its potential for advancing the field of semantic segmentation.

## 1. Introduction

The problem addressed in this work is the enhancement of semantic segmentation performance, with a specific focus on improving segmentation accuracy around mask boundaries. Semantic segmentation is a fundamental computer vision task that involves assigning a class label to each pixel in an image, effectively segmenting the image into meaningful regions corresponding to different object categories. While significant progress has been made in semantic segmentation using deep-learning models and advanced architectures, accurately segmenting object boundaries remains challenging. Often overlooked, object boundaries are critical for accurately delineating object shapes and providing precise segmentation results, but they are inherently ambiguous and can suffer from misclassifications [1].

One notable approach that addresses the challenge of improving segmentation qualities around boundaries is GSCNN [2]. GSCNN focuses on integrating the multi-task learning of binary boundaries for improving semantic segmentation using a Two-Stream approach. By considering both pixel-level semantics and boundary information, GSCNN leverages the global context to enhance the understanding of object structures, leading to more accurate segmentation results, especially around mask boundaries. Several researchers [3,4,5] have followed GSCNN by proposing various architectures to improve semantic segmentation.

Another relevant method is SegFix [6], a model-agnostic post-processing technique that aims to refine the segmentation results by explicitly addressing boundary inconsistencies. SegFix targets the problem of refining noisy boundaries by employing a novel post-processing model that is trained separately and iteratively refines the initial segmentation output.

In this context, we present the Semantic-Boundary-Conditioned Backbone (SBCB) framework, a simple but effective solution aimed at enhancing segmentation performance, particularly around mask boundaries. The core hypothesis guiding our work is that leveraging the complementary nature of semantic boundary detection (SBD) as an auxiliary task within the SBCB framework will significantly improve segmentation performance across various segmentation architectures without necessitating post-processing modules or specific architectural modifications. To achieve this, we incorporate the SBD task as an auxiliary component within a multi-task learning approach, as depicted in Figure 1.

The primary objective of the SBCB framework is to harness additional information from semantic boundaries during the training process. During training, we introduce a lightweight SBD head to the segmentation model’s backbone, training it jointly with the main segmentation task. This SBD head leverages multi-scale features from the backbone, allowing it to learn low-level features in the early stages and understand high-level semantics in the later stages. By exploiting the complementary nature of the SBD task, which focuses on discerning boundaries between different object categories, we effectively enhance the segmentation backbone’s representational capabilities, as demonstrated in Figure 2. We achieve this without requiring major architectural alterations or introducing additional computational overhead during inference. The contributions of this work are summarized below:*Semantic boundary detection as an auxiliary task*: We introduce the Semantic-Boundary- Conditioned Backbone (SBCB) framework, a novel training approach for semantic segmentation. It leverages semantic boundaries as an auxiliary task to enhance segmentation models without adding computational overhead during inference.*Seamless integration into a wide range of architectures*: The SBCB framework is flexible and can be applied to various existing backbones, such as ResNet and HRNet. We provide guidelines and implementations on customized architectures, such as BiSeNet, STDC, and the recent vision transformers.*Efficient data preprocessing during training*: The SBCB utilizes an on-the-fly ground-truth generation algorithm for semantic boundaries called the OTFGT module, which is efficient and compatible with commonly used training dataloaders.*Extensive benchmarks*: Along with the SBCB framework, we introduce the Binary-Boundary-Conditioned Backbone (BBCB) framework as a comparison method, aiming to explore the impact of different auxiliary tasks. Furthermore, extensive experiments validate the SBCB framework’s effectiveness in improving IoU, the boundary F-score, and over- and under-segmentation measures.*Potential directions for further studies*: Our research investigates novel explicit methods for leveraging the features obtained from the SBD head employed in the SBCB framework to facilitate feature fusion. We demonstrate the SBCB framework’s potential contributions to advancing research in multi-task models for semantic segmentation and SBD.*Accessibility and transparency*: The SBCB framework is open-source to benefit the community (source code: https://github.com/haruishi43/boundary_boost_mmseg).

By addressing this problem, we aim to advance the state of the art in semantic segmentation, particularly in scenarios where accurate boundary delineation is crucial, such as urban scenes, medical imaging, and remote sensing. For example, precise object segmentation masks can significantly benefit various downstream applications, such as object proposal generation [7], depth estimation [8], and image localization [9]. The proposed SBCB framework is expected to yield improved segmentation results, leading to more precise object delineation and overall better performance on challenging datasets like Cityscapes.

## 2. Related Work

### 2.1. Semantic Segmentation

In computer vision, semantic segmentation stands out as one of the most prominent and challenging tasks, leading to a diverse array of prior works aimed at addressing this problem. Long et al. [10] proposed an influential approach by introducing an end-to-end trainable fully convolutional network, which was adapted from image classification models for semantic segmentation. To capture multi-scale contextual information, Chen et al. [11] introduced dilated convolutions and atrous spatial pyramid pooling (ASPP) in their work. Another line of research by Zhao et al. [12] involved the creation of a pyramid-pooling module (PPM) to model multi-scale contexts. They also utilized an auxiliary FCN head to ensure stable training. Various methods [13,14,15,16,17,18,19] have been introduced to enhance the recognition of both local and global contexts by incorporating non-local operators [20] and self-attention mechanisms [21]. More recently, the adoption of vision transformers [22,23] for semantic segmentation has gained popularity, primarily due to their capacity to learn long-range contexts [24,25]. Moreover, there has been a notable surge of interest in universal image segmentation models [26,27,28,29] and interactive segmentation models [30,31,32], facilitated by the increasing popularity of larger models and datasets.

### 2.2. Edge and Semantic Boundary Detection

Similar to semantic segmentation, edge and boundary detection has been widely studied. Xie et al. [33] introduced a CNN model that can be trained end-to-end, which paved the way for various edge detection models, like those in [34,35]. Yu et al. [36] extended the task of binary edge detection to semantic boundary detection (SBD) by formulating the problem as multi-label pixel-wise classification. Hu et al. [37] introduced a dynamic fusion model with adaptive weights for better contextual modeling. Liu et al. [38] proposed DDS, a deep supervision framework that supervises all side outputs and is currently the state-of-the-art method for SBD.

### 2.3. Boundary-Aware Semantic Segmentation

There are many approaches to incorporate edges and boundaries into semantic segmentation, such as multi-task learning (MTL) [2,3,4,5,39,40,41], boundary-aware architectures [42,43], boundary-aware loss functions [44,45,46], and post-processing modules [6,47].

A widely used approach for boundary-aware segmentation is to apply MTL [48,49,50,51] to explicitly model edges and segmentation masks by jointly optimizing the network for the two tasks. In MTL, it is common to use a multi-head architecture with a shared backbone for memory efficiency. The backbone aims to learn a shared representation between the tasks, but this often fails due to the backbone being designed for a single task, leading to worse results [48,49]. Takikawa et al. [2] introduced an MTL framework using binary boundary detection as an auxiliary task to improve semantic segmentation, especially for pixels near mask boundaries. Similarly, Li et al. [3] introduced a novel framework for explicitly modeling the body and edge features. Graph representation has become increasingly popular for boundary-aware semantic segmentation, as proposed in [39,40]. Recently, ref. [41] proposed a lightweight encoder–decoder architecture utilizing boundary detection as auxiliary task.

Zhen et al. [4] introduced the first joint semantic segmentation and semantic boundary detection (JSB) model and proposed the iterative pyramid context module and duality loss, which enforces consistency between the two tasks. Yu et al. [5] proposed a complex dynamic graph propagation approach to couple the two tasks and refine segmentation and boundary maps.

SegFix [6] is a model-agnostic post-processing network that refines the output of a segmentation model with an independent network. The key idea of this method is to replace unreliable predictions in the mask boundaries with reliable interior labels. The post-processing network requires separate training.

Recently, there have been some creative approaches, like PIDNet [52] and SDN [53]. PIDNet addresses the overshoot phenomenon observed in segmentation networks by employing boundary guiding. SDN introduces a boundary-aware segmentation model by formulating boundary feature enhancement as an anisotropic diffusion process.

### 2.4. Positioning of Our Approach

Our approach is centered around developing a boundary-aware semantic segmentation system through the utilization of multi-task learning for semantic boundaries. In this work, we propose a novel training framework that conditions the segmentation model’s backbone by integrating well-established semantic boundary detection heads [36,37,38].

An essential aspect of our approach is that conditioning the backbone alone allows us to achieve competitive performance. This sets our method apart from existing joint models for semantic segmentation and boundary detection [2,4,5,41], which often require intricate modeling and complex feature fusion. Notably, our approach is designed as a single-step process that does not introduce additional parameters during inference. This is in contrast to post-processing methods like SegFix [6], which necessitates extra training and inference steps. Furthermore, our research focuses on a versatile training technique that can be applied across a wide range of existing segmentation models, distinguishing it from prior studies with more specific scopes.

## 3. Approach

The Semantic-Boundary-Conditioned Backbone (SBCB) framework, as illustrated in Figure 1, enhances semantic segmentation by incorporating a semantic boundary detection (SBD) head into the backbone network during training. This SBD head receives multi-scale features from selected stages of the backbone and is supervised using ground-truth (GT) semantic boundaries, which are dynamically generated on the fly using GT segmentation masks. Remarkably, during inference, when the task does not require semantic boundary information, the SBD head can be omitted, resulting in a semantic segmentation model without an increase in parameters.

To thoroughly introduce the SBCB framework, our paper proceeds with a systematic approach in the following sections. In Section 3.1, we comprehensively review existing SBD architectures while introducing the specific SBD heads utilized in our experiments. Moving forward, in Section 3.2, we delve into the details of the framework by applying the SBCB approach to two prominent backbone networks, namely, DeepLabV3+ and HRNet. In Section 3.3, we elucidate the on-the-fly (OTF) semantic boundary generation module, which is a pivotal component that endows this framework with remarkable flexibility and ease of use. Finally, in Section 3.4, we expound upon the loss function employed within the SBCB framework, which plays a crucial role in effectively optimizing and training the model.

### 3.1. Semantic Boundary Detection Heads

This section presents an overview of significant SBD models based on Convolutional Neural Networks (CNNs) that have emerged over the years. Understanding these SBD heads is crucial for comprehending their application in the SBCB framework and the conducted experiments. Additionally, we highlight some effective modifications that we have made during our reimplementation. Furthermore, we introduce the “Generalized” versions of these SBD heads, which are utilized in the SBCB framework.

**CASENet.** The CASENet architecture [36], proposed by Yu et al., presents a novel nested design without deep supervision on ResNet [54]. The architecture, shown in Figure 3b, modifies the ResNet backbone to capture higher-resolution features (detailed in Section 5.7). In each stage of the backbone (excluding stage 4), the features are passed into the Side Layer, comprising a 1×1 convolutional kernel, followed by a deconvolutional layer, increasing the resolution to match the input image. Throughout this paper, we interchangeably use the terms “Stage” and “Side”. Stages are based on the original backbone papers, often excluding the Stem. However, we use “Side”, a term used in the SBD-related literature, which includes the Stem. The last Side Layer (Side 5) produces an $N_{cat}\times H\times W$ tensor, while the other Side Layers (Sides 1 to 4) generate 1×H×W outputs, where $N_{cat}$ is the number of categories, and *H* and *W* are the height and width of the image. The outputs of the Side Layers are then processed by a Fuse Layer, which performs sliced concatenation of each feature, resulting in a $(4\times N_{cat})\times H\times W$ feature. This feature is further processed by a 1×1 convolution kernel to produce an $N_{cat}\times H\times W$ logit, supervised using the ground-truth semantic boundaries. Additionally, the output of the last Side Layer is also supervised with the same ground-truth semantic boundaries, acting as an auxiliary signal. Further details on the semantic boundary supervision loss $L_{\mathrm{SBD}}$ for the Fuse Layer and the last Side Layer are explained in Section 3.4. 

**Figure 3 sensors-23-06980-f003:**
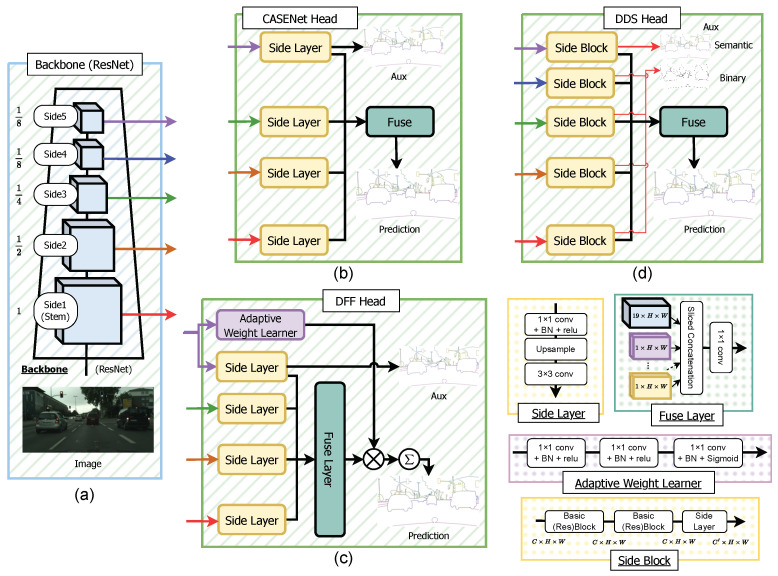
Overview of popular semantic boundary detection (SBD) architectures: The backbone produces multi-level features, as shown in (**a**). CASENet (**b**) utilizes Sides 1, 2, 3, and 5 of the backbone with modified Side Layers, employing a 1×1 convolutional kernel and bilinear upsampling with a 3×3 convolutional kernel. The features are concatenated and processed by a grouped 1×1 convolutional kernel. DFF (**c**) enhances CASENet by adding the Adaptive Weight Learner, which learns attentive weights applied to Fuse Layer outputs. DDS (**d**) expands on CASENet, incorporating Side 4 features and a deeper Side Block, supervised using a deep supervision method with binary boundaries for all Side Blocks. The auxiliary output from the last Side Block is also supervised with semantic boundaries.

In our implementation, we observed checkerboard artifacts in the original Side Layer outputs. To address this, we replaced the Side Layers with bilinear upsampling, followed by a 3×3 convolutional kernel, as depicted in Figure 3. This modification was adapted from techniques introduced for generative models using deconvolution [55], and we ensured that it did not increase the number of parameters.

**DFF.** The DFF architecture, introduced in [37], enhances the CASENet model by incorporating the Adaptive Weight Learner. This addition refines the output of the Fuse Layer using attentive weights. As depicted in Figure 3c, the Fuse Layer produces sliced concatenated features. Instead of using a standard 1×1 convolutional kernel, the Adaptive Weight Learner calculates weights, which are then applied to the tensor and summed to produce an output tensor of size $N_{cat}\times H\times W$. **DDS.** DDS [38] is the latest method that surpasses CASENet and DFF. It introduces a deeper Side Block composed of two ResNet Basic Blocks followed by a Side Layer. Figure 3d shows an overview of the network. Unlike CASENet, DDS explicitly supervises all Side Blocks, with the final output supervised by semantic boundaries and earlier outputs supervised by binary boundaries. **Generalized SBD heads.** To enable seamless integration within the SBCB framework, we introduce a generalized SBD head that can be applied to diverse backbone networks and segmentation architectures. This SBD head, referred to as the Generalized SBD head, is illustrated in Figure 4. Within our framework, we achieve this generalization by incorporating flexible Side and Fuse Layers, accommodating any of the previously mentioned SBD heads (CASENet, DFF, and DDS). The Side Layer can be adapted from CASENet’s Side Layers or DDS’s Side Blocks, while the Fuse Layer can take the form of either CASENet’s Fuse Layer or DFF’s Fuse Layer with the Adaptive Weight Learner. Moreover, our approach allows for the manipulation of the number of Sides, offering versatility to the framework. Specifically, in DDS, the *N*th side output is supervised using semantic boundaries, while binary boundaries supervise the earlier side outputs. This adaptability empowers the Generalized SBD head to seamlessly integrate with different segmentation architectures and backbones, providing enhanced flexibility for the SBCB framework. 

As a formal definition, the features obtained from the *k*th stage backbone are $S_{k}$, and the features obtained from the *k*th Side Layer are $B_{k}$. $S_{\mathrm{SBD}}$ represents a set of semantic boundary predictions, and $S_{\mathrm{Bin}}$ represents a set of binary boundary predictions. For CASENet and DFF, $S_{\mathrm{SBD}}=\left\{B_{N},B_{\mathrm{fuse}}\right\}$, where $B_{N}$ represents the last side output, and $B_{\mathrm{fuse}}$ represents the final fused prediction, as shown in Figure 4. For DDS, we supervise $S_{\mathrm{SBD}}=\left\{B_{N},B_{\mathrm{fuse}}\right\}$ and $S_{\mathrm{Bin}}=\left\{B_{k},\ldots B_{2},B_{1}\right\}$. Concretely, the Generalized SBD head can be defined as follows:(1)$$\begin{array}{cc} \left\{S_{N},\ldots ,S_{k},\ldots ,S_{1}\right\} & =\mathrm{Backbone}(I) \end{array}$$(2)$$\begin{array}{cc} B_{k} & ={\mathrm{SideLayer}}_{k}\left(S_{k}\right) \end{array}$$(3)$$\begin{array}{cc} B_{\mathrm{fuse}} & =\mathrm{Fuse}(\left\{B_{N},\ldots ,B_{k},\ldots ,B_{1}\right\}), \end{array}$$
where I is the input image.

### 3.2. SBCB Framework

In this section, we will demonstrate the application of the SBD heads reviewed in Section 3.1 within the SBCB framework. As mentioned earlier in Section 3, the SBCB framework incorporates an SBD head into the backbone, with each multi-scale feature directed to different Side Layers. Certain crucial factors need to be considered to ensure the ease of implementation across various backbones.

**Is the feature with the largest resolution passed to the first Side Layer?** To capture boundary details effectively, the first Side Layer must receive features with the largest resolution. Hence, we follow the SBD architecture convention and utilize the backbone’s stem if possible ($B_{1}$). Generally, a feature resolution of 1 or 1/2 of the input resolution suffices for the first Side Layer. **Which backbone features should be passed to which Side Layer?** When applying the SBCB to hierarchical backbones like ResNet, earlier stages ($B_{1}∼B_{N-1}$) are best-suited for the Binary-Side Layers, while the last stage ($B_{N}$) naturally fits the Semantic-Side Layer. Fortunately, most semantic segmentation architectures utilize hierarchical backbones, making the application of the SBCB framework straightforward. In cases such as HRNet, where features are hierarchical and branching out, it is crucial to incorporate all the features, typically by concatenating them. **Do the Side Layers receive features with sufficient resolution?** Although semantic segmentation models generally work with higher input resolutions, some backbones may reduce feature resolution excessively. In such cases, it is beneficial to adjust the convolutional kernel’s strides and dilations to increase the feature resolution. The goal is to ensure that the first side feature has a resolution of at least 1/2 of the input image. This technique, known as the “backbone trick”, is discussed in detail in Section 5.7. 

To enhance the comprehensiveness of the framework, we will present case studies demonstrating the application of the SBCB framework to popular architectures like DeepLabV3+ and HRNet. Furthermore, we will showcase how the SBCB framework can be seamlessly applied to other architectures, including those with heavily customized backbones. The detailed implementation and results of these case studies are provided in Section 6.

**DeepLabV3+ + SBCB**. Figure 5a illustrates the CASENet head applied to DeepLabV3+. The architecture follows a similar design to the SBD architectures with ResNet, and we utilize the “backbone trick” when dealing with small input image sizes. Implementing various SBD heads on DeepLabV3+ is generally straightforward, and we can readily incorporate the DDS head by incorporating Side 4 features and replacing the Side Layers with Side Blocks. **HRNet + SBCB.** The HRNet backbone consists of four stages, as depicted in Figure 5b. Since the first stage already reduces the resolution to 1/4, we utilize the features from the stem for the first Side Layer. Unlike ResNet, HRNet maintains consistent feature resolutions across its stages while branching out into smaller resolutions in each stage. To effectively incorporate these features, we resize and concatenate the features from each stage before passing them through the Side Layer. By including all the features from each stage, we aim to achieve improved conditioning of the backbone for enhanced performance. 

### 3.3. On-the-Fly Ground-Truth Generation

For the SBD and edge detection tasks, boundaries are manually annotated by human annotators. In some datasets, like Cityscapes, automatic preprocessing scripts are provided to generate GT boundaries from semantic and instance masks. These boundaries are generated before training and remain unchanged during training. On the other hand, for semantic segmentation tasks, it is a common practice to resize and rescale the GT masks during training to mitigate overfitting and introduce variations to the dataset. However, resizing the boundaries can result in inconsistent boundary widths, as illustrated in Figure 6. This will lead to the model learning inconsistent boundary widths, which is undesirable.

To address this issue, we developed a straightforward semantic boundary generation algorithm called the on-the-fly (OTF) semantic boundary GT generation module (OTFGT), as illustrated in Figure 7. This module takes a GT semantic segmentation mask $M^{\mathrm{GT}}$ as input and produces a semantic boundary mask $B^{\mathrm{GT}}$. For each category c∈C, where $M_{c}^{\mathrm{GT}}$ is a binary 2D array representing the category, we calculate a binary 2D array of boundaries $B_{c}^{\mathrm{GT}}$ using the equation:(4)$$B_{c}^{\mathrm{GT}}={\mathrm{Thresh}}_{r}\left(\mathrm{DT}\left(M_{c}^{\mathrm{GT}}\right)+\mathrm{DT}\left(1-M_{c}^{\mathrm{GT}}\right)\right).$$

Here, DT is a Euclidean distance transform function that computes the $L_{2}$ norm for each binary pixel using the method in [56]. We obtain the outer distance with $\mathrm{DT}(M_{c}^{\mathrm{GT}})$ and the inner distance with $\mathrm{DT}(1-M_{c}^{\mathrm{GT}})$. We then add the two distances to acquire the distances from the mask boundaries. The ${\mathrm{Thresh}}_{r}$ function thresholds the distance based on the radius *r* with the following condition:(5)$${\mathrm{Thresh}}_{r}\left(d\right)=\left\{\begin{array}{cc} 1 & d(i,j)\leq r\\ 0 & d(i,j)>r \end{array}\right..$$

We repeat the algorithm to generate the semantic boundaries of all categories *C*. For further details and Python code snippets, please refer to Appendix A.

### 3.4. Loss Functions

The model generates segmentation and boundary maps with pre-defined semantic categories from an input image. For the segmentation map, we apply cross-entropy (CE) loss, denoted by $L_{\mathrm{Seg}}$, to each pixel. As for the SBD head, binary cross-entropy (BCE) loss, $L_{\mathrm{SBD}}$, is applied for multi-label boundaries $S_{\mathrm{SBD}}$, following the approach in [36]. While CASENet and DFF utilize only multi-label boundaries for supervision, DDS introduces the deep supervision of edges by supervising earlier side outputs ($S_{\mathrm{Bin}}$) with binary boundary maps using BCE loss, $L_{\mathrm{Bdry}}$ [38].

The overall loss function used is defined as:(6)$$L=L_{\mathrm{Seg}}+\alpha \sum^{S_{\mathrm{SBD}}}L_{\mathrm{SBD}}+\beta \sum^{S_{\mathrm{Bin}}}L_{\mathrm{Bdry}},$$
where α and β are constants that balance the effects of losses from each task. We use the GT boundaries obtained from OTFGT ($B^{\mathrm{GT}}$) to supervise the boundaries.

## 4. Experimental Setup

### 4.1. Datasets

In our experiments, we utilized four datasets: Cityscapes, BDD100K, Synthia, and ADE20K datasets. We visualize the datasets in Figure 8, and the explanation of each dataset is provided below:**Cityscapes.** The Cityscapes dataset [57] is a widely used benchmark for semantic segmentation and semantic boundary detection. It contains 2975 training images, 500 validation images, and 1525 testing images, with annotations for 19 semantic categories. This dataset serves as the standard benchmark for evaluating SBD methods [36,37,38,58], and we conducted quantitative studies for both semantic segmentation and SBD on the validation set, while we benchmarked our method on the test set for semantic segmentation. **BDD100K.** The BDD100K dataset [59] was designed for multi-task learning in autonomous driving scenarios. It is the largest driving video dataset, comprising 100,000 video frames with annotations for ten tasks, including semantic segmentation. For our experiments, we used 10,000 images with a resolution of 1280×720, split into 7000 training and 1000 validation images. The dataset’s annotated labels align with those of the Cityscapes dataset. **Synthia.** The Synthia dataset [60] is a computer graphics dataset generated using a simulator, serving as an auxiliary dataset for Cityscapes and domain adaptation experiments. We utilized the “Rand” set of the dataset, containing 13,400 images with a resolution of 1280×760 and annotations for semantic categories matching Cityscapes. We explored the effect of the SBCB framework with precise boundary annotations using this stand-alone dataset, dividing it into 10,400 training, 1500 validation, and 1500 test images. **ADE20K.** The ADE20K dataset [61] is a scene-parsing dataset with 150 fine-grained semantic concepts. It consists of 20,210 training and 2000 validation images. Notably, this dataset’s domain is entirely different from previous street scene datasets, posing a challenge for evaluation. 

### 4.2. Evaluation Metrics

**Segmentation Metrics.** We evaluated the segmentation performance using the mean of intersection over union (mIoU). To assess the segmentation performance around the boundaries of the masks, we adopted the boundary F-score, following the approach in [2]. Unless explicitly stated, we used a pixel width of 3 px for the boundary F-score. 

Additionally, we employed the region-wise over-segmentation measure (ROM) and region-wise under-segmentation measure (RUM) recently proposed in [62]. ROM and RUM enable us to quantitatively measure the over- and under-segmentation characteristics of the models, providing a more objective evaluation compared to previous qualitative assessments. The values of ROM and RUM fall within the range of [0,1), where a value of 0 indicates no over- or under-segmentation, while higher values indicate increased over- or under-segmentation.

**Boundary Detection Metrics.** We followed [58] and adopted the maximum F-score (mF) at the optimal dataset scale (ODS) evaluated on the instance-sensitive “thin” protocol for SBD. 

### 4.3. Implementation Details

**Data Loading.** For the Cityscapes dataset, unless specified otherwise, we used a unified training crop size of 512 × 1024, 40 k training iterations, and a batch size of 8. For the Synthia, BDD100K, and ADE20K datasets, we used a crop size of 640×640. During ablation studies, discussed in Section 5, we trained for 80 k iterations on Synthia but reduced this to 40 k iterations for experiments described in Section 6 due to resource limitations. We further fine-tuned the models evaluated in the Cityscapes test benchmark for an additional 40 k iterations using the training and validation split, following the approach in [5]. Common data augmentations, such as random scaling (scale factors in [0.5,2.0]), horizontal flip, and photo-metric distortions, were applied. **Optimization.** During training, we used the SGD optimizer with a momentum coefficient of 0.9 and a weight decay coefficient of $5\times 10^{-4}$. The learning rate policy follows the “poly” learning rate decay, where the initial learning rate of 0.01 is multiplied by ${\left(1-\frac{\mathrm{iter}}{\max\_\mathrm{iter}}\right)}^{\gamma }$ with γ=9. **Loss.** For our loss function in Equation (Equation 6), we set α=5 and β=1. **Inference.** For the Cityscapes dataset, we conducted evaluations with single-scale whole inference. For the Synthia and BDD100K datasets, slide inference was performed. In Section 6.3, we evaluate semantic segmentation performance using multi-scale and flip (MS + Flip) inference with scales of [0.5,0.75,1.0,1.25,1.5,1.75,2.0]. **Software and Hardware.** All experiments were conducted using PyTorch, and the popular semantic segmentation framework “mmsegmentation” [63] was modified for our task. We ensured consistent software and hardware configurations for all experimental results. The models were trained on two NVIDIA A6000 GPUs and evaluated on a single NVIDIA RTX8000. 

## 5. Ablation Studies

In this section, we describe the ablation studies conducted to analyze the impact of the SBCB framework from various perspectives. In Section 5.1, we compare different SBD heads and select the most suitable candidate for experimentation throughout the paper. In Section 5.2, we explore the optimal side configuration to achieve the best performance with the SBCB framework. In Section 5.3, we examine which semantic categories benefit the most from the SBCB framework. In Section 5.4, we compare the SBCB framework with other auxiliary tasks to assess its effectiveness. In Section 5.5 and Section 5.6, we compare the SBCB framework with state-of-the-art multi-task and post-processing methods, demonstrating its ability to complement these methods and further improve segmentation quality. In Section 5.7, we investigate the effects of a simple yet effective modification to the backbone configuration, which improves segmentation and SBD performance. In Section 5.8, we analyze the effects of the SBCB framework on the SBD task. Finally, in Section 5.9 and Section 5.10, we demonstrate that our framework enhances segmentation around boundaries and addresses over- and under-segmentation issues through the boundary F-score and region-wise over-/under-segmentation measures (ROM and RUM) evaluations.

### 5.1. Which SBCB Head to Use?

In this section, we investigate the effects of using different semantic boundary detection (SBD) heads for the SBCB framework and determine the most suitable candidate for further evaluation.

Table 1 presents the performance of the DeepLabV3+ model trained using three different SBD heads, namely, CASENet, DFF, and DDS, in comparison with single-task baseline models. All SBD heads integrated into the SBCB framework demonstrate improvements over the single-task DeepLabV3+ model. Joint training also contributes to enhancing the SBD metric (maximum F-score). Additionally, we provide information on the number of parameters and computational costs (GFLOPs) introduced during training by the SBD heads. DDS incurs higher costs, but it is the most effective of the three heads. However, the trade-off of using DDS over CASENet for the SBCB framework may not be advantageous in terms of performance gains, particularly when evaluating DDS on other datasets and backbones.

Figure 9 showcases the qualitative results of applying the CASENet head to DeepLabV3+ compared to the baseline models. The additional semantic boundary supervision enables the model to detect small, thin objects more effectively. Furthermore, the SBCB framework enhances boundary detection by reducing artifacts and improving object perception. These results demonstrate the potential benefits of using SBD heads in the SBCB framework for semantic segmentation tasks.

**Different crop size.** We tested the SBD heads on a crop size of 769×769, another popular crop size in semantic segmentation. The results, shown in Table 2, exhibit a similar trend to the results in Table 1, with the CASENet head performing favorably. **Different backbone.** We evaluated the effects of using the HRNet-48 (HR48) backbone, and the results are displayed in Table 3. In this case, the CASENet head outperforms DDS and DFF by significant margins (1.0% and 0.5%, respectively). The CASENet head achieves an mF of 78.9%, identical to the heavy and inefficient single-task DDS model. **Different datasets.** As performance can vary across datasets, we further evaluated the SBD heads on the BDD100K dataset and Synthia, as shown in Table 4 and Table 5, respectively. On the BDD100K dataset, the DDS head significantly outperforms the baseline model and CASENet head. The DFF head also performs better than the CASENet head for this dataset for the first time. On Synthia, however, the CASENet head performs better than DDS. **Choice of SBD head: CASENet.** Considering the additional parameters and computational costs, using the CASENet head proves to be beneficial. Moreover, the SBD head in the SBCB framework is only used as an auxiliary signal, and the CASENet head outperforms DDS in some results. While the DDS head may produce higher metrics when computational costs are not a concern, for the rest of the analyses in this paper, we used the CASENet head as our primary SBD head for the SBCB framework. 

In Figure 2, we present qualitative visualizations comparing DeepLabV3+ with and without the CASENet head. The feature maps obtained from the last stage of the backbone conditioned on SBD show boundary-aware characteristics, resulting in reduced segmentation errors, especially around the boundaries. This demonstrates the effectiveness of the SBCB framework in improving the segmentation performance by incorporating boundary information during the feature extraction process.

### 5.2. Which Sides to Supervise?

Table 6 presents the effect of using different side configurations for the CASENet head applied to the ResNet backbone. The original configuration, which includes Sides 1, 2, 3, and 5, performs the best on two models (PSPNet and DeepLabV3). However, on DeepLabV3+, the configuration 1 + 2 + 3 + 4 + 5 outperforms the original configuration by 0.2%. While the performance gains between configurations are negligible, it is worth noting that each model may have an optimal side configuration within the SBCB framework. For fairness and consistency, we chose the original configuration (Sides 1, 2, 3, and 5) to evaluate other models and benchmark our methods for further evaluation. Users of the SBCB framework should be aware that different models might benefit from different side configurations.

### 5.3. Does It Improve All Categories?

Table 7 presents the per-category IoU comparisons for each model. While the SBCB framework generally improves most categories, some categories exhibit worse IoU scores. In particular, the categories “truck”, “bus”, and “train” are more affected, likely due to their relatively low number of samples and potential confusion with the “car” category. To address this, additional techniques, such as Online Hard Example Mining (OHEM), could be employed during training to focus on difficult samples and improve the overall performance on challenging categories with irregular boundaries and low sample counts.

From the table, it is evident that the SBCB framework significantly enhances the segmentation performance for categories characterized by complex shapes, such as “vegetation”, “terrain”, “person”, and “bike”. The incorporation of boundary-aware features into the backbone contributes to the segmentation head’s improved capability in accurately predicting the boundaries of these categories, consequently leading to superior segmentation results. This observation is visually apparent in Figure 2, where the segmentation errors surrounding the edges are notably reduced compared to the baseline DeepLabV3+ model, and it is particularly evident in scenarios such as people riding bikes in the bottom row.

One of the challenges in semantic segmentation arises from the presence of overlapping classes, which can occur due to occlusion or see-through objects. When occlusion is present, the predictions near the point of occlusion often become uncertain. This is evident in Figure 9, where objects like “poles” occlude a considerable portion of a building. The baseline model exhibits fragmentation in the prediction of the thin pole, as the larger wall provides more certainty in its segmentation. In contrast, the SBCB framework yields a cleaner segmentation mask, thanks to its ability to comprehend object boundaries more effectively.

Similarly, in scenarios involving see-through categories, such as the fence overlapping with a building in the same figure, the model faces a challenging task in making accurate predictions. However, the SBCB framework proves beneficial by generating a cleaner segmentation mask, leveraging its proficiency in understanding object boundaries more robustly.

The Cityscapes dataset contains annotations with inherent noise, particularly pronounced in categories featuring irregular boundaries, which inherently pose challenges to the segmentation task. This noise in annotations could potentially contribute to the observed variations in IoU scores across categories. Nevertheless, our belief is that the SBCB framework can still enhance segmentation results, even in the presence of noisy annotations, as long as the generated GT boundaries offer valuable guidance during the model’s training process. Although detecting such improvements purely based on quantitative metrics like IoU may be difficult, the qualitative outcomes presented in Figure 2, along with additional metrics like ROM and RUM (as discussed in Section 5.10), allow us to discern reduced fragmentation and enhanced accuracy in the segmentation masks.

### 5.4. Comparisons of Different Auxiliary Signals

The authors who introduced PSPNet [12] added an FCN head to the fourth stage (the one before the last stage) to the backbone to stabilize training and improve segmentation metrics. The auxiliary FCN head is trained on the same segmentation task as the main head and has been widely adopted in open-source projects such as mmseg.

Although not commonly used, various papers have explored using binary edge and boundary detection as an auxiliary task for semantic segmentation. Despite the difference in tasks, it has been found that the learned features in the edge detection head can be fused into the segmentation head.

In this section, we compare the SBCB framework with the mentioned auxiliary techniques, namely, “FCN” and “Binary Boundary Conditioned Backbone (BBCB)”. Note that BBCB is the SBCB framework applied to binary boundary detection. We applied the FCN, BBCB, and SBCB to three popular segmentation heads (PSPNet, DeepLabV3, and DeepLabV3+) with ResNet-101 as the backbone. The results on the Cityscapes validation split are shown in Table 8.

While all auxiliary signals improve IoU, the models trained using the SBCB framework consistently perform the best. The improvements obtained with the SBCB compared to the BBCB are around twofold, demonstrating the importance of the SBD task. The FCN shows significant gains of 0.7% when applied to PSPNet, but the FCN has a minimal impact on the other models. Both the BBCB and SBCB complement the FCN, achieving higher IoU results. Additionally, it is essential to consider the additional parameters introduced by these auxiliary signals during training. While the SBCB and BBCB only add thousands of parameters, the FCN adds 2.37 M parameters. Considering the performance gains and the additional parameters, it is evident that boundary-based auxiliary signals offer more benefits than the FCN.

We also evaluated the same models and auxiliary heads on the Synthia dataset, as shown in Table 9. Surprisingly, the FCN and BBCB do not provide significant performance gains and even have worse metrics than the baselines. However, the SBCB improves upon the baseline by over 1%. It is plausible that the features learned using the FCN could have conflicted with the main heads. In contrast to the noisy annotations in Cityscapes, Synthia contains precise segmentation masks rendered from a CG engine instead of human annotations. The classes “human” and “bike” in Synthia have small and thin segmentation masks, which adds to the difficulty. Although the features learned by the FCN complemented the main head features in Cityscapes, it appears that the FCN learned to derive a conflicted segmentation map for Synthia. The larger number of parameters in the FCN compared to the SBCB or BBCB might have contributed to this issue. Surprisingly, the BBCB did not perform as well as expected because it focuses on low-level features without explicitly modeling high-level semantics.

The SBCB framework conditions the backbone with SBD, a challenging task that focuses on low-level features while requiring high-level features for accurate boundary detection. The hierarchical modeling of the SBD task in the SBCB framework leads to better improvement in segmentation metrics compared to using the FCN or binary boundaries as auxiliary signals.

### 5.5. Comparisons with SegFix

In Table 10, we compare our framework with SegFix [6], a popular post-processing method. We obtained the results for SegFix by using the open-source code, which refines the output prediction based on the offsets learned using HRNet2x. Comparing the methods side-by-side, models trained with the SBCB framework, SegFix performs around 0.1%∼0.4% better than the SBCB. However, when combining the SBCB framework with the FCN (as mentioned in Section 5.4), we achieve competitive performance and significantly outperform SegFix on two models.

It is important to consider that SegFix is an independent post-processing model, while our framework produces competitive results without any post-processing or additional parameters during inference. SegFix adds a post-processing module that requires separate training. Furthermore, SegFix is specifically designed to correct predictions around mask boundaries, which can be challenging for the base model to predict accurately. As a result, the base model might not actively learn boundary-aware features. In contrast, our training framework conditions the backbone to be boundary-aware by solving SBD, as demonstrated in Section 5.9. In other words, SegFix and our framework are complementary because boundary-aware predictions are easier for SegFix to correct. This is evident from the significant improvements achieved by using the SBCB along with SegFix, as shown in the table.

### 5.6. Comparisons with GSCNN

GSCNN [2] is a well-known semantic segmentation model that incorporates a binary boundary detection multi-task architecture with a dedicated boundary detection head, called the shape stream, branching out from the Side Layers, similar to the SBD heads in the SBCB framework. The main distinction is that GSCNN explicitly merges the features from the shape stream into the semantic segmentation head. GSCNN, based on the ResNet-101 backbone, is a customized version of DeepLabV3+ that utilizes an ASPP module.

While it may be challenging to make a direct apples-to-apples comparison due to the different loss functions and the explicit feature merging in GSCNN, we aimed to evaluate how effectively the SBCB framework enhances DeepLabV3+ in comparison to different configurations of GSCNN. Table 11 presents the results of our comparison. The baseline GSCNN is GSCNN without the image gradient (Canny Edge). “+Canny” is the original configuration with the image gradient. We also experimented with supervising the shape stream using the SBD task, denoted by “SBD”, and modified the shape stream by increasing the channels. Finally, we used the SBCB framework on GSCNN, denoted by “+SBCB”, which adds the SBD head to the backbone without any other modifications.

Comparing DeepLabV3+ with GSCNN, we observe a substantial improvement of +1.0% when using GSCNN. The SBD supervision on GSCNN results in a slightly lower improvement of +0.5%, indicating that boundary signals do have a positive impact on semantic segmentation. The reduction in improvement can be attributed to the Gated convolution kernel, which restricts features to a single channel, leading to a degradation in representation capability.

However, the SBCB framework proves to be highly effective in improving DeepLabV3+. When utilizing CASENet and DDS as SBD heads, the SBCB framework achieves improvements of +0.8% and +1.1%, respectively. These improvements are competitive with the results obtained using the original GSCNN configuration, further emphasizing the potential of the SBCB framework.

The flexibility of the SBCB framework allows it to be easily applied to GSCNN as well, leading to an even higher improvement of +1.4%. This demonstrates the versatility and efficacy of the SBCB framework, which can enhance segmentation performance across different models.

### 5.7. Backbone Trick

In this section, we explore the use of the “backbone trick”, which is a modification to the backbone architecture introduced to obtain better edge detection and semantic boundary detection (SBD) performance. The “backbone trick” involves modifying the strides and dilations of the backbone stages to increase the output resolutions without changing the number of parameters, making it suitable for edge detection and SBD tasks.

In edge detection and SBD tasks, higher-resolution feature maps are essential to accurately capture small edges and boundaries. Traditional backbones like ResNet, designed for image classification, produce smaller feature maps that may not be well suited for edge detection. By applying the “backbone trick”, we can retain the pre-trained weights while achieving higher-resolution feature maps, improving edge detection and SBD performance.

Similarly, in semantic segmentation, we modified the strides and dilations of the last two stages to maintain the final feature resolution at 1/8 of the input image size, which is commonly used for accurate segmentation. The configurations of the two modifications are shown in Table 12.

Table 13, Table 14 and Table 15 present results using the HED version of ResNet-101 (HED ResNet-101) on the Cityscapes, BDD100K, and Synthia datasets, respectively. Compared to the normal segmentation ResNet-101 in Table 1, Table 4, and Table 5, HED ResNet-101 generally achieves better performance for both single-task and SBCB framework models. The Synthia dataset, in particular, shows higher performance gains, benefiting from the higher-resolution feature maps that capture more detailed and precise ground truths.

While the “backbone trick” is commonly applied to ResNet-101, it can also be extended to other backbones, such as transformer backbones, as demonstrated in Section 6.7. By conditioning the backbones with SBD through the SBCB framework, we can achieve significant performance improvements without complex modeling, making it a practical and effective approach for enhancing edge detection and SBD tasks.

### 5.8. Does SBCB Also Improve SBD Metrics?

Based on the previous ablations studies, it is clear that the SBCB framework improves the metrics for semantic segmentation. In addition to semantic segmentation, we also evaluated the performance of the models trained using the SBCB framework on semantic boundary detection (SBD) tasks, as shown in Table 16.

The results demonstrate that models trained with the SBCB framework achieve significant improvements in SBD performance compared to state-of-the-art single-task methods. The improvements range from 5% to over 10%, showcasing the effectiveness of the SBCB framework in enhancing boundary detection. Moreover, when comparing our DeepLabV3+ model trained with the SBCB framework to the joint semantic segmentation and semantic boundary detection model CSEL, our method outperforms CSEL without explicitly utilizing the features learned in the segmentation head with feature fusion. This demonstrates that the SBCB framework, which is primarily designed for semantic segmentation, effectively improves SBD performance as well.

Overall, the SBCB framework’s success can be attributed to its ability to condition the backbone for semantic segmentation tasks, resulting in improved performance for both semantic segmentation and semantic boundary detection without the need for complex modeling explicitly dedicated to boundary detection.

### 5.9. Does SBCB Improve Segmentation around Boundaries?

In Table 17, we present the boundary F-scores for both baseline models and models trained with the SBCB framework. The results clearly indicate that the models trained with the SBCB framework consistently achieve higher boundary F-scores, particularly when the trimap widths are smaller. The improved boundary F-scores demonstrate the effectiveness of the SBCB framework in enhancing the model’s ability to accurately detect and delineate object boundaries. We believe that the SBCB framework enables the backbone to learn and preserve boundary-aware features, which results in segmentation masks with better quality around the mask boundaries.

### 5.10. Does SBCB Improve Over- and Under-Segmentation?

In this section, we evaluate the effects of the SBCB framework in terms of over- and under-segmentation using the recently proposed region-based over-segmentation measure (ROM) and region-based under-segmentation measure (RUM) [62]. The results are presented in Table 18, where lower ROM and RUM values indicate better segmentation quality, reflecting reduced over- and under-segmentation, respectively.

The table clearly shows that the models trained using the SBCB framework consistently exhibit improvements in both ROM and RUM metrics. This indicates that the SBCB framework effectively mitigates over- and under-segmentation issues in the segmentation outputs. Semantic boundary conditioning in the SBCB framework reinforces strict distinction in object groupings, which helps to resolve unwanted partitioning and leads to an overall improvement in the segmentation quality.

For detailed per-category results of ROM and RUM, please refer to Appendix B. Furthermore, the qualitative analysis in Figure 9 provides visual evidence of the improved segmentation around the boundaries. For instance, the over-segmentation of the pole is notably reduced by the application of the SBCB framework.

While the improvements in under-segmentation may not be as easily distinguishable in qualitative comparisons, the quantitative evaluation using ROM and RUM metrics confirms the effectiveness of the SBCB framework in addressing both over- and under-segmentation issues in semantic segmentation tasks.

## 6. Experiments

In this section, we present a comprehensive evaluation of the proposed Semantic-Boundary-Conditioned Boosting (SBCB) approach. The evaluation was conducted by applying it to various architectures and datasets, aiming to assess its impact on semantic segmentation performance.

We begin by exploring the effectiveness of SBCB training across a wide range of backbone architectures and popular segmentation heads in Section 6.1 and Section 6.2. Next, we benchmark our method with the DeepLabV3+ architecture on the Cityscapes dataset in Section 6.3. We compare the results with state-of-the-art (SOTA) methods to demonstrate the superiority of our approach. In Section 6.4, we present experiments on the challenging ADE20k dataset. Furthermore, we provide the results of SBCB training on recent lightweight segmentation architectures in Section 6.5 and Section 6.6 to showcase the flexibility and effectiveness of the SBCB framework. Finally, we validate the compatibility of the SBCB training paradigm with modern backbones, ConvNeXt and Segformer, in Section 6.7. This evaluation underscores the continued relevance and applicability of our approach in the evolving landscape of semantic segmentation.

### 6.1. Different Backbones

Table 19 and Table 20 present the performance improvements achieved by employing the SBCB framework during the training of various backbones. Notably, we evaluated our approach on two datasets with different levels of annotation qualities, demonstrating the robustness and consistency of the SBCB framework.

Our findings in both tables reveal that the SBCB framework consistently leads to significant improvements in intersection over union (IoU), even across different backbone architectures. In particular, the Cityscapes evaluation showcases enhancements in F-score, ROM, and RUM metrics for every backbone, illustrating the effectiveness of our method.

Furthermore, we provide qualitative results of the SBCB framework on the Cityscapes dataset in Appendix C, further substantiating the impact and practicality of our approach. These results collectively underscore the potential of the SBCB in boosting semantic segmentation performance across diverse scenarios.

### 6.2. Different Heads

In Table 21 and Table 22 we present the performance evaluation of models trained with the SBCB framework, where we utilized different segmentation heads while keeping the backbone fixed at ResNet-101.

The results in these tables demonstrate the consistent improvement achieved by the SBCB framework in terms of intersection over union (IoU) and the boundary F-score across various segmentation heads. Notably, the IoU metric is consistently enhanced for all the examined heads. ROM is improved for every segmentation head, while RUM is improved for every head except for OCR. However, OCR has the most gains in IoU and the boundary F-score, leading us to believe that this is a trade-off in performance.

To complement our quantitative findings, we include qualitative results of the SBCB framework on the Cityscapes dataset in Appendix C. These visual results further substantiate the efficacy of our proposed approach and showcase the learned features from the last stage of the backbone.

### 6.3. Cityscapes Benchmark

**Cityscapes Validation Split.** Table 23 presents the performance of DeepLabV3+ trained using the SBCB framework, along with a comparison to other SOTA methods with and without boundary auxiliary training. Notably, our SBCB-empowered DeepLabV3+ surpasses other methods in performance while leveraging an off-the-shelf segmentation head and backbone, without the need for explicit architecture redesign. This underscores the effectiveness of our approach in boosting semantic segmentation performance without significant modifications to the base model. 

Furthermore, the SBCB-empowered DeepLabV3+ achieves competitive results compared to joint-task models, which are inherently designed to better incorporate boundary information into their architectures.

These findings highlight the remarkable capabilities of the SBCB framework in enhancing semantic segmentation, offering a compelling alternative to achieve state-of-the-art results without complex architectural changes.

**Cityscapes Benchmark.** Table 24 displays the performance of DeepLabV3+ trained using the SBCB framework and provides a comparison with other SOTA models on the Cityscapes Benchmark. While our approach did not surpass the performance of SOTA multi-task methods, DeepLabV3+ trained with the SBCB framework demonstrated competitive performance and was able to match the results of some SOTA models. These results underscore the effectiveness of the SBCB framework in enhancing the performance of DeepLabV3+ on the challenging Cityscapes dataset, positioning it as a compelling alternative in the landscape of semantic segmentation methods. While not outperforming all SOTA models, our approach exhibits valuable competitiveness, indicating that performance gains can be achieved with boundary conditioning. 

### 6.4. Experiments on ADE20k

We trained DeepLabV3+ models using both ResNet-50 and ResNet-101 as backbones and carefully compared their performance against models trained using the SBCB framework on the challenging ADE20k dataset. The compelling results of these experiments are presented in Table 25, where it is evident that the SBCB framework yields notable improvements of over 0.5% compared to the base models.

### 6.5. BiSeNet

In our pursuit of broader applicability and performance gains, we extended the application of the SBCB framework to the Bilateral Segmentation Network (BiSeNet) V1 and V2, which are specialized models designed for real-time semantic segmentation [66,67].

Both BiSeNet V1 and V2 architectures comprise a split backbone, consisting of the Detail Path (or Spatial Path) and the Semantic Path (or Context Path). The Detail Path is a shallow CNN with a few stages, retaining large feature resolutions (four stages for BiSeNet V1 and three stages for BiSeNet V2). The Semantic Path, on the other hand, is a deeper CNN tailored to capture high-level semantics. While BiSeNet V1 adopts off-the-shelf architectures like ResNet-50 for the Semantic Path, BiSeNet V2 employs a customized six-stage ConvNet with FCN auxiliary heads for supervising features from the middle stages.

To incorporate the SBCB framework, we selected specific stages (Sides) of the backbone to be supervised by the SBD head. Specifically, we chose three stages from the Detail Path as the Binary Sides for the SBD head and the last stage of the Semantic Path as the Semantic Side. It is essential to highlight that our approach did not require any modifications to the original model. We simply added the SBD head by extracting the mid-features from the backbones. For more details, please refer to Appendix D.

The results obtained through the SBCB framework on BiSeNet (V1 and V2) are presented in Table 26. As anticipated, applying the SBCB framework led to improvements in both IoU and boundary F-score metrics, further confirming its effectiveness in enhancing models based on non-conventional architectures. This outcome underscores the versatility and potential performance gains offered by the SBCB framework, even for specialized models like BiSeNet V1 and V2, thereby contributing to the advancement of semantic segmentation research in real-time scenarios.

### 6.6. STDC

Like BiSeNet, the STDC network is efficient for real-time semantic segmentation [68]. However, the STDC network is a single-branch network that replaces the Detail Path with the Detail Head, which uses the features from the third stage to perform “detail guidance” only during the training phase. The Detail Head is supervised with “Detail GT”, which is generated using a multi-scale Laplacian convolution kernel in an on-the-fly manner, similar to our method. The Detail GT contains spatial details like boundaries and corners.

In the following experiment, we replaced the Detail Head with the SBD head and trained using the SBCB framework. We took the first four stages of the backbone as the Binary Sides and used the output of the FFM as the Semantic Side for the SBD head. Please see Appendix E for more details.

The results are shown in Table 26, where we compare the original STDC with STDC that replaced the Detail Head with our SBD head. Remarkably, substantial improvements are observed when employing the SBD head as the auxiliary task, particularly in terms of IoU metrics. While the Detail Head aims to enhance the segmentation quality around boundaries, our SBCB framework demonstrates higher improvements in the boundary F-score, further accentuating its efficacy in leveraging boundary information to enhance semantic segmentation.

### 6.7. ConvNeXt and SegFormer

In this section, we explore the applicability of the SBCB framework and the “backbone trick” on two contemporary architectures: ConvNeXt and SegFormer.

ConvNeXt represents a backbone architecture composed of pure ConvNet components with design elements borrowed from vision transformers (ViT) [22,69]. On the other hand, SegFormer is an architecture designed for segmentation, featuring a ViT-based backbone called the Mix Transformer (MiT), along with a lightweight All-MLP segmentation head [25]. Notably, both architectures incorporate hierarchical feature extraction, rendering them compatible with the SBCB framework.

In Table 27, we present the results obtained by applying the SBCB framework to these modern architectures. Additionally, we assess the impact of integrating the “backbone trick”, denoted by “Mod”, into the backbones.

The results demonstrate that the SBCB framework remains effective in improving these modern state-of-the-art architectures, leading to consistent performance gains in terms of both IoU and the boundary F-score. This outcome further confirms the versatility of the SBCB approach and its ability to enhance the performance of contemporary models by leveraging boundary information, reinforcing its relevance in advancing the field of semantic segmentation.

## 7. Feature Fusion

In addition to our primary focus on the Semantic-Boundary-Conditioned Boosting (SBCB) framework, we also explored the explicit utilization of features obtained from the semantic boundary head through two feature fusion techniques. These techniques aim to further enhance segmentation performance by leveraging the knowledge learned in the SBD head.

**Channel-Merge.** The first technique, known as Channel-Merge, involves straightforward channel concatenation with a few convolutional kernels to facilitate feature fusion, as depicted in Figure 10a. In this method, we take the features before upsampling from the Side Layers of the SBD head. Each feature is resized and concatenated into a single tensor, which is further combined with the features obtained from the segmentation head (e.g., pyramid-pooling module (PPM)). To integrate the features effectively in the channel direction, we employ two 1×1 convolutional kernels. It is worth noting that the number of convolutions can be adjusted according to specific requirements.**Two-Stream Merge.** The second technique, known as Two-Stream Merge, establishes a direct connection between the features learned in the SBD head and the segmentation head, achieved by employing a 1×1 convolutional kernel, as illustrated in Figure 10b. This approach is inspired by the GSCNN architecture, wherein we treat the SBD head as the shape stream, and the fusion mechanism mirrors that of the GSCNN.

While these feature fusion techniques are not the primary focus of this paper, they serve as valuable supplementary approaches to leverage the knowledge acquired in the SBD head for improved segmentation performance. By explicitly incorporating boundary-related information through these fusion methods, we seek to further enhance the segmentation quality and offer additional insights into the potential benefits of integrating boundary-aware features into the segmentation process.

Table 28 and Table 29 presents the results of two baseline architectures, where we applied the SBCB framework and feature fusion methods. The evaluation was performed on two datasets: Cityscapes and Synthia.

While we initially expected the Two-Stream architecture to exhibit superior performance, we observed that, while it performed well on certain datasets, it was outperformed by the SBCB framework on some occasions. Surprisingly, the Channel-Merge architecture achieved the best IoU metrics across all models and datasets, and it also displayed the best boundary F-scores in most cases.

Comparing the SBCB framework with the Channel-Merge approach against the baseline model, we noticed a significant improvement in segmentation performance when using the SBCB framework. This highlights the substantial contributions of the SBCB framework, primarily driven by the representational capabilities of the backbone, while the feature fusion methods yielded smaller improvements on top of the improvements obtained with the SBCB framework. This may also highlight the importance of the boundary-aware feature representations learned by the backbone.

Furthermore, we observed that the Channel-Merge method proved particularly advantageous when the training ground-truth masks were precise. The Synthia dataset, which provides generated segmentation masks, benefits more from the feature fusion approach due to its cleaner annotations, whereas the Cityscapes dataset naturally contains noisier boundaries attributed to human annotators.

It is important to note that feature fusion methods introduce a dependency of the segmentation head on the SBD head, which in turn increases computational costs. This dependency also introduces the complexity of designing the fusion methods, which must be carefully tuned to avoid instability. For example, Channel-Merge might be better than Two-Stream Merge due to the Merge module mixing various representations obtained earlier rather than a single representation obtained at the end of the SBD head.

The SBCB framework remains instrumental in consistently enhancing existing segmentation models without modifications to the original architecture. The insights gleaned from the SBD heads hold promise for inspiring novel joint architectures, such as Channel-Merge and Two-Stream Merge.

## 8. Conclusions

We present the SBCB framework, a compelling and straightforward training approach that effectively enhances segmentation performance. At its core, the framework incorporates a semantic boundary detection (SBD) head, which is applied to the hierarchical features of the backbone and supervised by semantic boundaries. Our exploration of different SBD heads revealed that the CASENet architecture significantly improves segmentation quality without introducing a substantial increase in parameters during training.

Through comprehensive experiments on popular backbones and segmentation heads, conducted on the challenging Cityscapes and Synthia datasets, we demonstrate the efficacy of the SBCB framework. It consistently improves segmentation quality while effectively addressing the challenges of over- and under-segmentation, particularly in regions near boundaries. The obtained results reveal significant average improvements, with the SBCB framework boosting IoU by 1.2%, the boundary F-score by 2.6%, ROM by 0.011, and RUM by 0.005 on the Cityscapes dataset.

Moreover, our explorations extend beyond conventional architectures, as we applied the SBCB framework to customized backbones and recent transformer-based architectures. The results illustrate the versatility of our approach, confirming its compatibility with various models and architectures.

In addition to its inherent effectiveness, we offer modifications and feature fusion methods to promote the broader utilization of semantic boundaries for semantic segmentation. The Channel-Merge technique, in particular, yields notable improvements, especially when dealing with precise training ground-truth masks.

Looking ahead, we envision applying the SBCB framework to other segmentation tasks, such as depth estimation, panoptic segmentation, universal image segmentation [28], and interactive segmentation models [30], aiming to influence and improve other dense prediction tasks. This framework can also be extended to support other tasks for auxiliary supervision such as depth, normal, and distance transforms. The success and effectiveness of the SBCB framework open up exciting possibilities for future research in semantic segmentation and related areas.

## Figures and Tables

**Figure 1 sensors-23-06980-f001:**
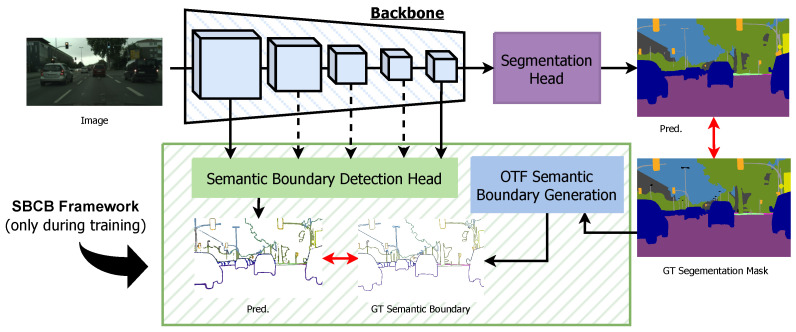
Overview of the Semantic-Boundary-Conditioned Backbone (SBCB) framework. During training, the semantic boundary detection (SBD) head is integrated into the backbone of the semantic segmentation head. Ground-truth (GT) semantic boundaries are generated on the fly (OTF) by the semantic boundary generation module to train the SBD head. This straightforward framework enhances segmentation quality by encouraging the backbone network to explicitly and jointly model boundaries and their relation with semantics, as the SBD task is complementary yet more challenging than the main task.

**Figure 2 sensors-23-06980-f002:**
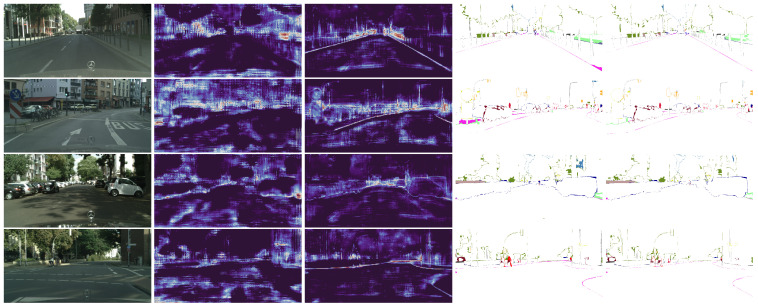
Visualization of the backbone features and segmentation errors of DeepLabV3+ with and without the SBCB framework. Starting from the left, the columns represent the input image, last-stage features without the SBCB, last-stage features with the SBCB, segmentation errors without the SBCB, and segmentation errors with the SBCB. Backbone features conditioned on semantic boundaries exhibit boundary-aware characteristics. Consequently, this results in better segmentation, especially around the mask boundaries. Best seen in color and zoomed in.

**Figure 4 sensors-23-06980-f004:**
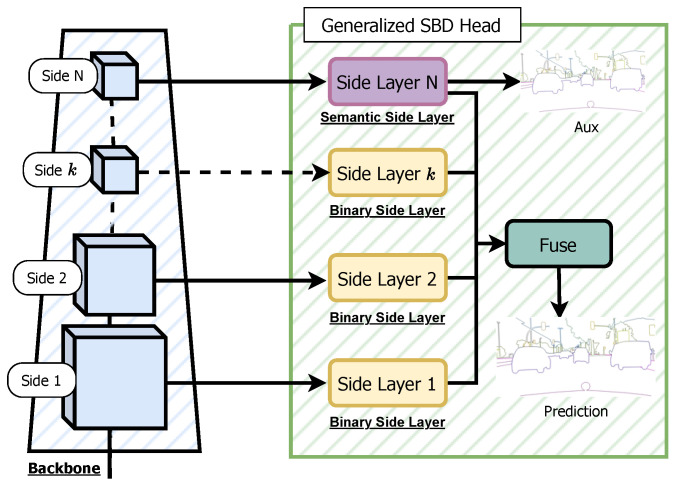
Overview of the Generalized SBD head. The Generalized CASENet Architecture is an extended version of the original CASENet architecture shown in Figure 3. In this generalized version, the last (*N*th) Side Layer is referred to as the Semantic-Side Layer, and its corresponding input feature is called the Semantic Side. Conversely, the 1∼(N−1)th Side Layer is termed the Binary-Side Layer, with the input side feature denoted as the Binary Side, having a single channel like other SBD architectures. This generalization allows for the flexibility of accommodating an unrestricted number of Sides and Side Layers, enabling the application of this SBD head to various backbone networks. Moreover, this generalization can be extended to work with DFF and DDS architectures.

**Figure 5 sensors-23-06980-f005:**
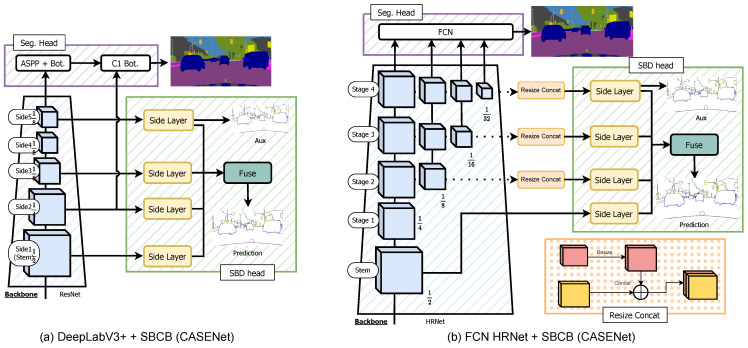
A diagram showcasing how the SBCB framework is applied to a DeepLabV3+ segmentation head is shown in (**a**). A diagram showcasing how the SBCB framework is applied to the HRNet backbone with an FCN segmentation head is shown in (**b**).

**Figure 6 sensors-23-06980-f006:**
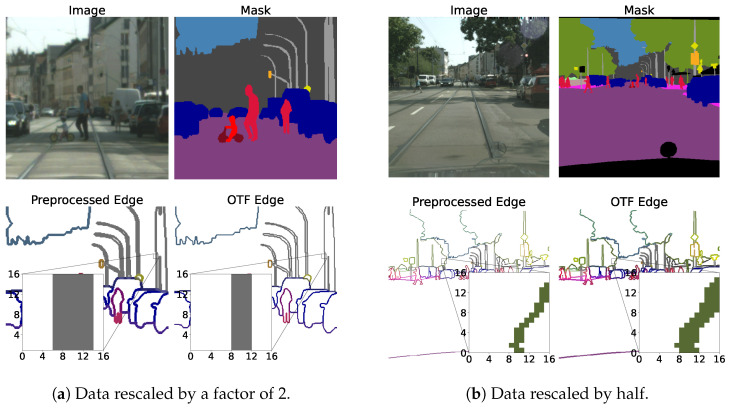
The two figures represent sample validation images, masks, and boundaries from the Cityscapes validation split, which we rescaled and cropped to 512×512. In each figure, we compare the two methods of preprocessing. The one on the left uses preprocessed boundaries, and the one on the right uses OTFGT boundaries. We can observe that OTFGT boundaries have consistent boundary widths, while preprocessed boundaries vary depending on the rescale value.

**Figure 7 sensors-23-06980-f007:**
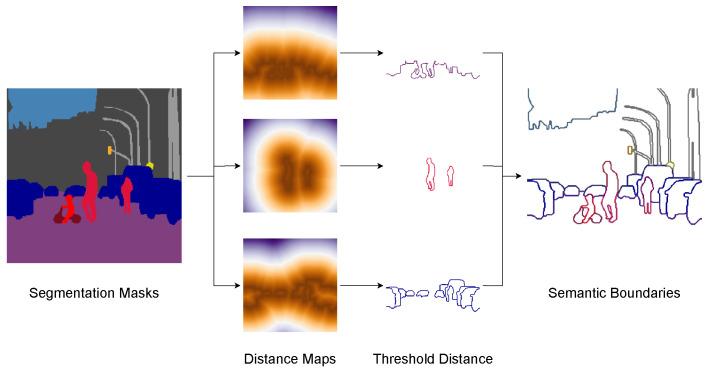
Overview of the OTFGT module. We apply distance transforms to segmentation masks to obtain category-specific distance maps. We then threshold the distances by the radius of the boundaries to obtain category-specific boundaries. The boundaries are concatenated to form a semantic boundary tensor for supervision.

**Figure 8 sensors-23-06980-f008:**
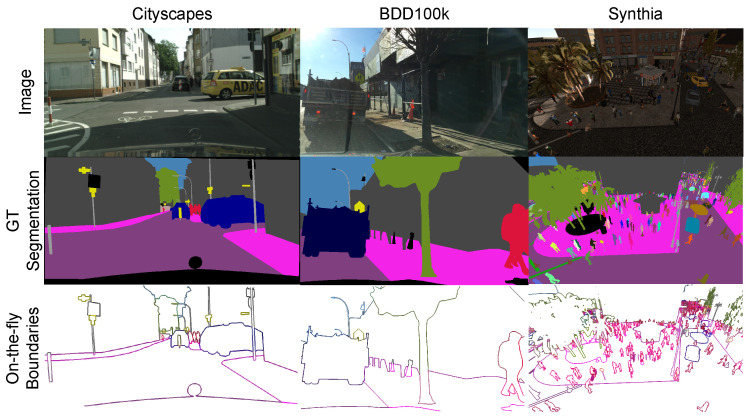
The three main datasets that we used for the experiments. We show a sample input image, segmentation GT, and the result of OTF semantic boundary generation for each dataset. The color of the segmentation masks and boundaries corresponds to the colors used in the Cityscapes dataset. For Synthia, the visualization contains separate colors for each mask instances. Humans annotated the Cityscapes and BDD100K datasets, and the segmentation masks are clean but tend to have imperfections around the boundaries and exhibit “polygon” masks. On the other hand, the Synthia dataset is data from a game engine, and the annotations are pixel-perfect, making this a challenging dataset for semantic segmentation. The segmentation mask for Synthia also contains instance segmentation, which is used for OTF semantic boundary generation but not for the segmentation task. The BDD100K and Synthia datasets are less widely used than the Cityscapes dataset. However, the BDD100K and Synthia datasets contain more variations in natural noise and corruption (weather, heavy light reflections, etc.), which will help benchmark the methods fairly. The images are best seen in color and zoomed in.

**Figure 9 sensors-23-06980-f009:**
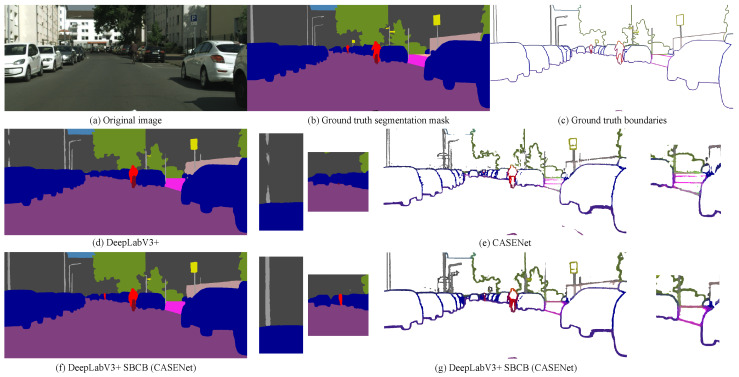
Visualizations of the GT and predictions on the Cityscapes dataset. Depicted in (**a**–**c**) are the input image, ground-truth (GT) segmentation map, and GT semantic boundary map. Note that because the task of SBD is pixel-wise multi-label classification, the visualized semantic boundary maps have overlapped boundaries. The color of the segmentation and boundaries represents categories following the visualization format used in Cityscapes. In (**d**), we show the output of DeepLabV3+, a popular semantic segmentation model. The semantic boundary detection (SBD) baseline is CASENet, which we show in (**e**). The output of DeepLabV3+ trained with the SBCB framework using the CASENet head is shown in (**f**,**g**). We can see that small and thin objects are recognized better using the framework and smoother boundaries with fewer artifacts. We can also notice improvements in over-segmentation for both the segmentation mask and semantic boundary results.

**Figure 10 sensors-23-06980-f010:**
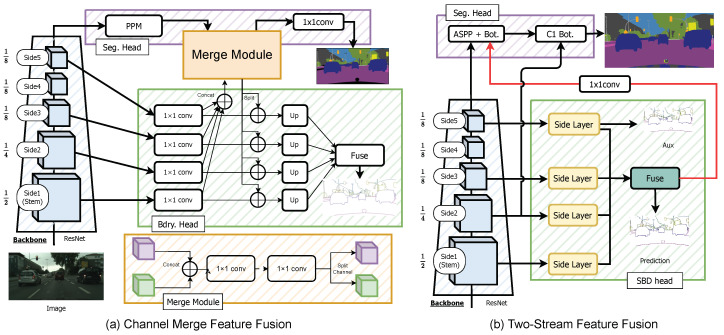
In (**a**), we show how to apply the Channel-Merge module for explicit feature fusion based on the SBCB framework. In (**b**), we show how to apply the Two-Stream approach for explicit feature fusion modeled after the GSCNN architecture.

**Table 1 sensors-23-06980-t001:** Results for Cityscapes. The best performant metrics are shown in bold.

Head	mIoU	mF (ODS)	Param.	GFLOPs
DeepLabV3+	79.5	-	60.2 M	506
CASENet	-	63.7	42.5 M	357
DFF		65.5	42.8 M	395
DDS		73.4	243.3 M	2079
SBCB (CASENet)	80.3	74.4	60.2 M	508
SBCB (DFF)	80.2	74.6	60.5 M	545
SBCB (DDS)	**80.6**	**75.8**	261.0 M	2228

**Table 2 sensors-23-06980-t002:** Results for Cityscapes with input crop size of 769×769. The best performant metrics are shown in bold.

Head	mIoU	mF (ODS)	Param.	GFLOPs
DeepLabV3+	78.9	–	60.2 M	506
CASENet	-	68.6	42.5 M	357
DFF		68.9	42.8 M	395
DDS		75.5	243.3 M	2079
SBCB (CASENet)	80.3	74.0	60.2 M	508
SBCB (DFF)	80.0	74.8	60.5 M	545
SBCB (DDS)	**80.4**	**75.6**	261.0 M	2228

**Table 3 sensors-23-06980-t003:** Results for Cityscapes using the HRNet-48 (HR48) backbone. The best performant metrics are shown in bold.

Head	mIoU	mF (ODS)	Param.	GFLOPs
FCN	80.5	–	65.9 M	187
CASENet	-	75.7	65.3 M	172
DFF		75.3	65.5 M	210
DDS		78.9	89.0 M	946
SBCB (CASENet)	**82.0**	78.9	65.9 M	187
SBCB (DFF)	81.5	78.8	66.0 M	221
SBCB (DDS)	81.0	**79.3**	89.5 M	1012

**Table 4 sensors-23-06980-t004:** Results for BDD100K. The best performant metrics are shown in bold.

Head	mIoU	mF (ODS)
DeepLabV3+	60.0	-
CASENet	-	55.7
DFF		57.3
DDS		59.9
SBCB (CASENet)	61.4	56.6
SBCB (DFF)	62.0	58.1
SBCB (DDS)	**64.1**	**60.2**

**Table 5 sensors-23-06980-t005:** Results for Synthia. The best performant metrics are shown in bold.

Head	mIoU	mF (ODS)
DeepLabV3+	74.5	-
CASENet	-	61.0
DFF		64.8
DDS		**67.6**
SBCB (CASENet)	**75.9**	65.2
SBCB (DFF)	75.3	66.5
SBCB (DDS)	75.7	67.0

**Table 6 sensors-23-06980-t006:** Results using ResNet-101 backbone with different side configurations on the Cityscapes validation split.

Head	Sides	mIoU	Δ
PSPNet		77.6	
	1 + 5	78.5	+0.9
	1 + 2 + 5	78.6	+1.0
	1 + 2 + 3 + 5	78.7	**+1.1**
	1 + 2 + 3 + 4 + 5	78.5	+0.9
DeepLabV3		79.2	
	1 + 5	79.8	+0.6
	1 + 2 + 5	79.9	**+0.7**
	1 + 2 + 3 + 5	79.9	**+0.7**
	1 + 2 + 3 + 4 + 5	79.4	+0.2
DeepLabV3+		79.5	
	1 + 5	80.1	+0.6
	1 + 2 + 5	80.1	+0.6
	1 + 2 + 3 + 5	80.3	+0.8
	1 + 2 + 3 + 4 + 5	80.5	**+1.0**

**Table 7 sensors-23-06980-t007:** Per-category IoU for the Cityscapes validation split. Red and Blue represents improvements and degradation.

Method	SBCB	mIoU	Road	Swalk	Build.	Wall	Fence	Pole	Tlight	Sign	Veg	Terrain	Sky	Person	Rider	Car	Truck	Bus	Train	Motor	Bike
PSPNet	✓	77.6	98.0	83.9	92.4	49.5	59.3	64.5	71.7	79.0	92.4	64.2	94.7	81.8	60.5	95.0	77.8	89.1	80.1	63.4	77.9
		78.7	98.3	85.7	92.7	52.7	60.7	66.3	72.7	80.8	92.8	64.3	94.6	82.4	62.7	95.3	79.5	88.6	81.4	66.0	78.7
		+1.1	+0.3	+1.8	+0.3	+3.2	+1.4	+1.8	+1.0	+1.8	+0.4	+0.1	−0.1	+0.6	+2.2	+0.3	+1.7	−0.5	+1.3	+2.6	+0.8
DeepLabV3	✓	79.2	98.1	84.6	92.6	54.5	61.7	64.6	71.7	79.3	92.6	64.6	94.6	82.4	63.8	95.4	83.2	90.9	84.2	67.7	78.1
		79.9	98.4	86.4	93.0	55.3	63.7	66.8	72.9	80.4	94.9	65.4	94.9	83.3	65.9	95.5	81.9	92.3	81.3	68.2	78.9
		+0.7	+0.3	+1.8	+0.4	+0.8	+2.0	+2.2	+1.2	+1.1	+2.3	+0.8	+0.3	+0.9	+2.1	+0.1	−1.3	+1.4	−2.9	+0.5	+0.8
DeepLabV3+	✓	79.5	98.1	85.0	92.9	53.2	62.8	66.5	72.1	80.4	92.7	64.9	94.7	82.8	63.6	95.5	85.1	90.9	82.2	69.4	78.4
		80.3	98.3	85.9	93.4	65.7	65.6	68.5	73.0	81.4	92.8	66.1	95.3	83.3	65.6	95.5	81.3	88.3	78.1	68.7	78.8
		+0.8	+0.2	+0.9	+0.5	+12.5	+2.8	+2.0	+0.9	+1.0	+0.1	+1.2	+0.6	+0.5	+2.0	0	−3.8	−2.6	−4.1	−0.7	+0.4

**Table 8 sensors-23-06980-t008:** Tables that compare different backbone-conditioning methods on the Cityscapes validation split. We investigated the effects on three popular segmentation heads: PSPNet, DeepLabV3, and DeepLabV3+. Note that all methods use ResNet-101 as the backbone. Also, note that we show the number of parameters (Params.) during training.

Head	FCN	BBCB	SBCB	Param.	mIoU	Δ
PSPNet				65.58 M	77.6	
	✓			+2.37 M	78.3	+0.7
		✓		+0.01 M	78.1	+0.5
			✓	+0.05 M	**78.7**	+1.1
	✓	✓		+2.37 M	79.1	+1.5
	✓		✓	+2.41 M	**79.4**	+1.8
DeepLabV3				84.72 M	79.2	
	✓			+2.37 M	79.3	+0.1
		✓		+0.01 M	79.6	+0.4
			✓	+0.05 M	**79.9**	+0.7
	✓	✓		+2.37 M	**80.1**	+0.9
	✓		✓	+2.41 M	**80.1**	+0.9
DeepLabV3+				60.2 M	79.5	
	✓			+2.37 M	79.7	+0.2
		✓		+0.01 M	79.9	+0.4
			✓	+0.05 M	**80.3**	+0.8
	✓	✓		+2.37 M	**80.6**	+1.1
	✓		✓	+2.41 M	80.5	+1.0

**Table 9 sensors-23-06980-t009:** Tables that compare different backbone-conditioning methods on Synthia.

Head	FCN	BBCB	SBCB	mIoU	Δ
PSPNet				70.5	
	✓			70.1	−0.4
		✓		70.7	+0.2
			✓	**71.7**	+1.2
	✓	✓		70.7	+0.2
	✓		✓	**71.6**	+1.1
DeepLabV3				70.9	
	✓			70.6	−0.3
		✓		70.7	−0.2
			✓	**71.9**	+1.0
	✓	✓		70.5	−0.4
	✓		✓	**71.0**	+0.1
DeepLabV3+				72.4	
	✓			72.0	−0.4
		✓		72.1	−0.3
			✓	**73.5**	+1.1
	✓	✓		72.3	−0.1
	✓		✓	**73.5**	+1.1

**Table 10 sensors-23-06980-t010:** Comparison of the use of SegFix with auxiliary heads (SBCB and FCN) on the Cityscapes validation split.

Model		mIoU	Δ
PSPNet		77.6	
	+ SegFix	78.8	+1.2
	+ SBCB	78.7	+1.1
	+ SBCB + FCN	79.4	+1.8
	+ SBCB + SegFix	79.7	+2.1
	+ SBCB + FCN + SegFix	80.3	+2.8
DeepLabV3		79.2	
	+ SegFix	80.3	+1.1
	+ SBCB	79.9	+0.7
	+ SBCB + FCN	80.1	+0.9
	+ SBCB + SegFix	80.8	+1.6
	+ SBCB + FCN + SegFix	81.0	+1.8
DeepLabV3+		79.5	
	+ SegFix	80.4	+0.9
	+ SBCB	80.3	+0.8
	+ SBCB + FCN	80.6	+1.1
	+ SBCB + SegFix	81.0	+1.5
	+ SBCB + FCN + SegFix	81.2	+1.7

**Table 11 sensors-23-06980-t011:** Comparisons of DeepLabV3+ and GSCNN on the Cityscapes validation split. We show that the SBCB framework can be applied to train GSCNN.

Model		mIoU	Δ
DeepLabV3+		79.5	
	+SBCB (CASENet)	80.3	+0.8
	+SBCB (DDS)	80.6	+1.1
GSCNN		80.5	+1.0
	+Canny	80.6	+1.1
	SBD	80.0	+0.5
	+SBCB (CASENet)	80.9	+1.4

**Table 12 sensors-23-06980-t012:** This table shows the configurations of the two common types of modifications to the ResNet backbone. Note that the output feature resolutions are in the order of Stem, Stages 1, Stage 2, Stage 3, and Stage 4.

Task	Stem Stride	Strides	Dilations	Resolutions
Original	2	(1, 2, 2, 2)	(1, 1, 1, 1)	(1/2, 1/4, 1/8, 1/16, 1/32)
Segmentation	2	(1, 2, 1, 1)	(1, 1, 2, 4)	(1/2, 1/4, 1/8, 1/8, 1/8)
Edge Det.	1	(1, 2, 2, 1)	(2, 2, 2, 4)	(1, 1/2, 1/4, 1/8, 1/8)

**Table 13 sensors-23-06980-t013:** Results of the “Backbone Trick” validated on Cityscapes. We modified the ResNet-101 backbone’s stride and dilation at each stage to keep the number of parameters the same but generate larger feature maps. The authors of [33] introduced this technique, and we prepend “HED” to the backbone that uses this trick.

Head	mIoU	mF (ODS)	Param.	GFLOPs
DeepLabV3+	79.8	-	60.2 M	506
CASENet	-	68.6	42.5 M	417
DFF		70.0	42.8 M	455
DDS		76.3	243.3 M	2661
SBCB (CASENet)	**81.0**	75.1	60.2 M	508
SBCB (DFF)	80.8	75.4	60.5 M	545
SBCB (DDS)	80.8	**76.5**	261.0 M	2228

**Table 14 sensors-23-06980-t014:** Results of the “Backbone Trick” validated on BDD100K.

Head	mIoU	mF (ODS)
DeepLabV3+	59.8	-
CASENet	-	56.6
DFF		58.1
DDS		60.1
SBCB (CASENet)	62.4	59.3
SBCB (DFF)	62.0	58.9
SBCB (DDS)	**63.5**	**60.5**

**Table 15 sensors-23-06980-t015:** Results of the “Backbone Trick” validated on Synthia.

Head	mIoU	mF (ODS)
DeepLabV3+	77.0	-
CASENet	-	64.0
DFF		65.6
DDS		68.5
SBCB (CASENet)	78.0	67.5
SBCB (DFF)	77.8	**68.9**
SBCB (DDS)	**78.6**	68.4

**Table 16 sensors-23-06980-t016:** Comparison of SBD models on the Cityscapes validation split using the instance-sensitive “thin” evaluation protocol. ^†^: Performance reported in [38].

Method	Backbone	mF (ODS)
CASENet ^†^	HED ResNet-101	68.1
SEAL ^†^	HED ResNet-101	69.1
STEAL ^†^	HED ResNet-101	69.7
DDS ^†^	HED ResNet-101	73.8
CSEL [5]	HED ResNet-101	78.1
DeepLabV3+ + SBCB (CASENet)	ResNet-101	77.8
DeepLabV3+ + SBCB (CASENet)	HED ResNet-101	78.4
DeepLabV3+ + SBCB (DDS)	ResNet-101	**78.8**
DeepLabV3+ + SBCB (DDS)	HED ResNet-101	**78.8**

**Table 17 sensors-23-06980-t017:** Comparison of the boundary F-score, evaluated on the Cityscapes validation split. The models were trained using the same hyperparameters and ResNet-101 backbone.

Head	SBCB	12 px	Δ	9 px	Δ	5 px	Δ	3 px	Δ
PSPNet		80.9		79.6		75.7		70.2	
	✓	83.3	+2.4	82.1	+2.5	78.5	+2.8	73.3	+3.1
DeepLabV3		81.8		80.6		76.7		71.2	
	✓	83.4	+1.6	82.2	+1.6	78.7	+2.0	73.4	+2.2
DeepLabV3+		81.2		80.0		76.4		71.4	
	✓	83.0	+1.8	81.8	+1.8	78.5	+2.1	73.7	+2.3

**Table 18 sensors-23-06980-t018:** Comparison of region-based over-segmentation measure (ROM) and region-based under-segmentation measure (RUM) on the Cityscapes validation split. The models are trained using the same hyperparameters and ResNet-101 backbone.

Head	SBCB	ROM ↓	Δ	RUM ↓	Δ
PSPNet		0.078		0.102	
	✓	0.061	−0.017	0.098	−0.004
DeepLabV3		0.072		0.104	
	✓	0.060	−0.012	0.1	−0.004
DeepLabV3+		0.08		0.094	
	✓	0.065	−0.015	0.086	−0.008

**Table 19 sensors-23-06980-t019:** Effect of using SBCB for different CNN-based backbones on Cityscapes.

Head	Backbone	SBCB	mIoU ↑	Δ	F-Score ↑	Δ	ROM ↓	Δ	RUM ↓	Δ
DenseASPP	ResNet-50		77.5		69.0		0.108		0.096	
		✓	78.3	+0.8	70.6	+1.6	0.1	−0.008	0.093	−0.003
DenseASPP	DenseNet-169		76.6		69.0		0.077		0.102	
		✓	78.2	+1.6	72.1	+3.1	0.072	−0.005	0.101	−0.001
ASPP	ResNeSt-101		79.5		72.3		0.079		0.102	
		✓	80.3	+0.8	75.2	+2.9	0.065	−0.014	0.094	−0.008
OCR	HR18		78.9		71.9		0.074		0.093	
		✓	79.7	+0.8	74.0	+2.1	0.066	−0.008	0.092	−0.001
OCR	HR48		80.7		74.4		0.073		0.09	
		✓	82.0	+1.3	77.7	+3.7	0.069	−0.004	0.083	−0.007
ASPP	MobileNetV2		73.9		66.2		0.074		0.1	
		✓	74.4	+0.5	68.3	+2.1	0.07	−0.004	0.095	−0.005
LRASPP	MobileNetV3		64.5		58.0		0.128		0.082	
		✓	67.5	+3.0	62.1	+4.1	0.115	−0.013	0.08	−0.002

**Table 20 sensors-23-06980-t020:** Effect of using SBCB for different CNN-based backbones on Synthia.

Head	Backbone	SBCB	mIoU ↑	Δ
DenseASPP	ResNet-50		69.6	
		✓	70.5	+0.9
DenseASPP	DenseNet-169		71.3	
		✓	72.0	+0.7
ASPP	ResNeSt-101		72.3	
		✓	73.8	+1.5
OCR	HR18		70.1	
		✓	70.9	+0.8
OCR	HR48		74.3	
		✓	76.0	+1.7
ASPP	MobileNetV2		65.3	
		✓	67.0	+1.7
LRASPP	MobileNetV3		60.8	
		✓	64.8	+4.0

**Table 21 sensors-23-06980-t021:** Effect of using SBCB with different segmentation heads on Cityscapes. Note that the backbones for all models are set to ResNet-101.

Head	SBCB	mIoU ↑	Δ	F-Score ↑	Δ	ROM ↓	Δ	RUM ↓	Δ
FCN		74.6		69.3		0.072		0.104	
	✓	76.3	+1.7	71.6	+2.3	0.058	−0.014	0.096	−0.008
PSPNet		77.6		70.2		0.078		0.102	
	✓	78.7	+1.1	73.3	+3.1	0.061	−0.017	0.098	−0.004
ANN		77.4		70.1		0.074		0.1	
	✓	79.0	+1.6	72.8	+2.7	0.059	−0.015	0.091	−0.009
GCNet		77.8		70.2		0.07		0.103	
	✓	78.9	+1.1	73.0	+2.8	0.058	−0.012	0.092	−0.011
ASPP		79.2		71.2		0.072		0.104	
	✓	79.9	+0.7	73.4	+2.2	0.06	−0.012	0.1	−0.004
DNLNet		78.7		71.2		0.07		0.101	
	✓	79.7	+1.0	73.6	+2.4	0.052	−0.018	0.093	−0.008
CCNet		79.2		71.9		0.068		0.102	
	✓	80.1	+0.9	73.9	+2.0	0.053	−0.015	0.089	−0.013
UPerNet		78.1		71.9		0.082		0.091	
	✓	78.9	+0.8	73.9	+2.0	0.068	−0.014	0.087	−0.004
OCR		78.2		70.6		0.071		0.096	
	✓	80.2	+2.0	74.4	+3.8	0.064	−0.007	0.1	+0.004

**Table 22 sensors-23-06980-t022:** Effect of using SBCB with different segmentation heads on Synthia. Note that the backbones for all models are set to ResNet-101.

Head	SBCB	mIoU	Δ
FCN		70.0	
	✓	70.9	+0.9
PSPNet		70.5	
	✓	71.7	+1.2
ANN		70.4	
	✓	71.8	+1.4
GCNet		70.8	
	✓	71.4	+0.6
ASPP		70.9	
	✓	71.9	+1.0
DNLNet		70.5	
	✓	71.9	+1.4
CCNet		70.8	
	✓	71.3	+0.5
UPerNet		72.4	
	✓	73.1	+0.7
OCR		69.7	
	✓	72.4	+2.7

**Table 23 sensors-23-06980-t023:** Comparison of our method and state-of-the-art methods on the Cityscapes validation split. The methods were only trained with fine-annotation data and without additional coarse training data and Mapillary Vistas pre-training. The sections are divided into three categories: models without boundary auxiliary, with boundary auxiliary, and with SBCB auxiliary.

Method	Backbone	mIoU
PSPNet [12]	ResNet-101	78.8
DeepLabV3+ [64]	ResNet-101	78.8
CCNet [16]	ResNet-101	80.5
DANet [13]	ResNet-101	81.5
SegFix [6]	ResNet-101	81.5
GSCNN [2]	ResNet-38	80.8
RPCNet [4]	ResNet-101	82.1
CSEL [5]	HED ResNet-101	83.7
BANet [41]	HED ResNet-101	82.5
DeepLabV3+ SBCB	ResNet-101	82.2
DeepLabV3+ SBCB	HED ResNet-101	82.6

**Table 24 sensors-23-06980-t024:** Comparison of our method and state-of-the-art methods on the Cityscapes test split. The methods were only trained with fine-annotation data and without additional coarse training data and Mapillary Vistas pre-training.

Method	Backbone	mIoU
PSPNet [12]	ResNet-101	78.4
PSANet [65]	ResNet-101	80.1
SeENet [15]	ResNet-101	81.2
ANNNet [14]	ResNet-101	81.3
CCNet [16]	ResNet-101	81.4
DANet [13]	ResNet-101	81.5
RPCNet [4]	ResNet-101	81.8
CSEL [5]	HED ResNet-101	82.1
DeepLabV3+ SBCB	ResNet-101	81.4
DeepLabV3+ SBCB	HED ResNet-101	81.0

**Table 25 sensors-23-06980-t025:** Results using ResNet backbones on the ADE20k validation split.

Head	Backbone	Batch	SBCB	mIoU	Δ
PSPNet	50	8		39.9	
	50	8	✓	40.6	+0.7
	101	4		38.2	
	101	4	✓	38.7	+0.5
DeepLabV3+	50	8		41.5	
	50	8	✓	42.0	+0.5
	101	4		37.7	
	101	4	✓	38.2	+0.5

**Table 26 sensors-23-06980-t026:** Results for BiSeNet and STDC on Cityscapes validation split.

Model	SBCB	mIoU	Δ	F-Score	Δ
BiSeNetV1 R50		74.3		66.0	
	✓	75.4	+1.1	69.9	+3.9
BiSeNetV2		70.7		63.8	
	✓	71.6	+0.9	66.2	+2.4
STDC V1 FCN (+Detail Head)		73.7		66.5	
STDC V1 FCN	✓	75.4	+1.7	67.9	+1.4

**Table 27 sensors-23-06980-t027:** Results of the SBCB framework on modern backbones/architectures on the Cityscapes validation split.

Head	Backbone	SBCB	mIoU	Δ	F-Score	Δ
	ConvNeXt-base		81.8		74.4	
		✓	82.0	+0.2	75.5	+1.1
UPerNet	Mod ConvNeXt-base	✓	82.2	+0.4	76.5	+2.1
	MiT-b0		75.5		66.9	
SegFormer		✓	76.5	+1.0	68.1	+1.2
	Mod MIT-b0	✓	76.8	+1.3	69.7	+2.8
	MiT-b2		80.9		73.2	
SegFormer		✓	81.1	+0.2	74.7	+1.5
	Mod MIT-b2	✓	81.6	+0.7	76.0	+2.8
SegFormer	MiT-b4		81.6		75.5	
		✓	82.2	+0.6	76.7	+1.2

**Table 28 sensors-23-06980-t028:** Comparison of feature fusion methods with baseline methods on Cityscapes.

Model		mIoU	Δ	F-Score	Δ
		77.6		70.2	
	+SBCB	78.7	+1.1	**73.3**	+3.1
PSPNet	Two-Stream Merge	78.7	+1.0	73.0	+2.8
	Channel-Merge	**79.1**	+1.5	73.2	+3.0
		79.5		71.4	
	+SBCB	80.2	+0.7	73.7	+2.3
DeepLabV3+	Two-Stream Merge	**80.5**	+1.0	73.6	+2.2
	Channel-Merge	**80.5**	+1.0	**74.5**	+3.1

**Table 29 sensors-23-06980-t029:** Comparison of feature fusion methods with baseline methods on Synthia.

Model		mIoU	Δ	F-Score	Δ
		70.5		63.7	
	+SBCB	71.7	+1.2	65.9	+2.2
PSPNet	Two-Stream Merge	71.3	+0.8	65.5	+1.8
	Channel-Merge	**72.5**	+2.0	**67.2**	+3.5
		72.4		67.2	
	+SBCB	73.5	+1.1	69.1	+1.9
DeepLabV3+	Two-Stream Merge	73.8	+1.4	69.2	+2.0
	Channel-Merge	**74.0**	+1.6	**69.9**	+2.7

## Data Availability

No new data were created.

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
