# Peer review of "Boosting Semantic Segmentation by Conditioning the Backbone with Semantic Boundaries"

_sensors, 2023, doi:10.3390/s23156980_

Round 1

Reviewer 1 Report

Incorporating semantic boundaries into the backbone architecture, the conditioning approach improves the performance of semantic segmentation models by enabling better boundary localization, enhanced discriminative features, improved handling of complex shapes, and increased robustness to noisy or incomplete boundaries. These advantages contribute to more accurate and visually appealing segmentation results in a wide range of applications. The paper has been written well.

Suggestions:

1. Fusion process aims to enhance the representation of the features related to the boundaries, making them more discriminative for semantic segmentation. So, feature fusion can be explained in detail for better clarity.

2. Explain how the conditioning approach helps in handling complex shapes and overlapping classes in semantic segmentation.

3. Can the conditioning approach handle noisy or incomplete boundaries? How does it ensure robustness in such cases?

4. What are some possible variations or extensions of the conditioning approach that can be explored in future research?

The advantages of the research work lie in its ability to improve segmentation accuracy, enhance boundary localization, handle complex scenes, adapt to different scenarios, handle noisy or incomplete boundaries, and provide a better understanding of semantic boundaries. The paper can be accepted after minor revision.

Author Response

Thank you for your valuable feedback on our paper. Please see the attachment.

Reviewer 2 Report

This paper proposes a multi-task framework that uses semantic boundary detection (SBD) as an auxiliary task. This is an interesting and practical study, but there are still some issues that need to be explained and corrected for further improvement. (1)  The abstract needs to be rewritten for highlighting the purpose and focus.

(2)  The author points out that this framework is model-agnostic. What does the “model-agnostic” mean? I do not think this is an omnipotent framework, and there is still necessary connections to the designed backbones.

(3)  In the section On-the-fly Ground Truth Generation, please provide detailed descriptions of its executive process and calculation formula.

(4)  Although the author has done a lot of work, the experimental section needs to be reorganized for better reading.

(5)  The language needs further improvement.

The English language needs further improvement.

Author Response

(The authors gave the same response as above.)

Reviewer 3 Report

Suggestion to be implemented before publication

1. Problem statement and hypothesis needs to be define more precisely.

2. Latest literature compative survey needs to be added.

3. Propsposed method with mathematical model is expected in this article in a more concise way,

Suggestion to be implemented before publication

1. Problem statement and hypothesis needs to be define more precisely.

2. Latest literature compative survey needs to be added.

3. Propsposed method with mathematical model is expected in this article in a more concise way,

Author Response

(The authors gave the same response as above.)
